# Generalizing Abstention for Noise-Robust Learning in Medical Image Segmentation

**Wesam Moustafa**[1,2]                                      WESAM.H.MOUSTAFA@GMAIL.COM
**Hossam Elsafty**[1,2,3] iD                                HOSSAM.ELSAFTY@IAIS.FRAUNHOFER.DE
**Helen Schneider**[1,2,3]                                  HELEN.SCHNEIDER@IAIS.FRAUNHOFER.DE
**Lorenz Sparrenberg**[1,3] iD                             SPARRENBERG@BIT.UNI-BONN.DE
**Rafet Sifa**[1,2,3]                                       RAFET.SIFA@IAIS.FRAUNHOFER.DE

[1] *Institute of Computer Science, University of Bonn, Bonn, Germany*

[2] *Fraunhofer Institute for Intelligent Analysis and Information Systems IAIS, Sankt Augustin, Germany*

[3] *Lamarr Institute for Machine Learning and Artificial Intelligence, Dortmund, Germany*

**Editors:** Accepted for publication at MIDL 2026

## Abstract

Label noise is a critical problem in medical image segmentation, often arising from the inherent difficulty of manual annotation. Models trained on noisy data are prone to overfitting, which degrades their generalization performance. While a number of methods and strategies have been proposed to mitigate noisy labels in the segmentation domain, this area remains largely under-explored. The abstention mechanism has proven effective in classification tasks by enhancing the capabilities of Cross Entropy, yet its potential in segmentation remains unverified. In this paper, we address this gap by introducing a universal and modular abstention framework capable of enhancing the noise-robustness of a diverse range of loss functions. Our framework improves upon prior work with two key components: an informed regularization term to guide abstention behaviour, and a more flexible power-law-based auto-tuning algorithm for the abstention penalty. We demonstrate the framework's versatility by systematically integrating it with three distinct loss functions to create three novel, noise-robust variants: GAC, SAC, and ADS. Experiments on the CaDIS and DSAD medical datasets show our methods consistently and significantly outperform their non-abstaining baselines, especially under high noise levels. This work establishes that enabling models to selectively ignore corrupted samples is a powerful and generalizable strategy for building more reliable segmentation models. Our code is publicly available at https://github.com/wemous/abstention-for-segmentation.

**Keywords:** Abstention, Medical Image Segmentation, Label Noise, Noise-Robust Learning, Loss Functions.

## 1. Introduction

The remarkable advancements in deep learning have revolutionized numerous fields, largely propelled by the availability of labelled datasets (Karimi et al., 2020; Zhang et al., 2020; Garcia-Garcia et al., 2017). However, the presence of label noise is a significant impediment to the generalizability of Deep Neural Networks (DNNs), as their immense capacity makes them prone to memorizing incorrect labels, which harms their ability to generalize to unseen data (Lienen and Hüllermeier, 2024; González-Santoyo et al., 2025; Schneider et al., 2024b, 2023). This problem is especially pronounced in medical image segmentation, where

obtaining clean, pixel-level annotations is notoriously difficult and expensive, and where annotation errors can have direct clinical consequences (Karimi et al., 2020; Zhang et al., 2020; Xu et al., 2024). Training on such noisy labels leads to incorrect gradients, causing the model to learn erroneous patterns and fail in critical applications (Marcinkiewicz and Mrukwa, 2019).

To counteract label noise, a variety of robust learning methodologies have been developed, primarily for classification tasks. These include noise filtering techniques (González-Santoyo et al., 2025), loss reweighting strategies (Karimi et al., 2020), and curriculum learning (Lienen and Hüllermeier, 2024). While promising, these methods often introduce computational complexity or require strong assumptions about the noise characteristics (González-Santoyo et al., 2025). Among the explored directions, noise-robust loss functions are a compelling alternative due to their simplicity, efficiency, and model-agnostic nature (Staats et al., 2025). By leveraging properties like boundedness (Zhang and Sabuncu, 2018) and symmetry (Wang et al., 2019), they modify the optimization objective to inherently limit the influence of noisy examples and prevent overfitting (Toner and Storkey, 2023; Ding et al., 2024).

Despite these advances, a research gap remains for robust learning specifically in image segmentation, where existing methods often struggle to address the spatially-correlated inaccuracies inherent in annotation noise (Guo et al., 2025; Karimi et al., 2020). In this paper, we propose to address this gap by adapting and generalizing the **abstention** mechanism, a powerful technique that has proven effective in mitigating label noise in classification (Karimi et al., 2020; Thulasidasan et al., 2019; Schneider et al., 2024a). The abstention mechanism empowers a DNN to abstain from making a prediction on confusing or unreliable samples by integrating an abstention option directly into the training process. Building upon the foundational Deep Abstaining Classifier (DAC) (Thulasidasan et al., 2019) and its extension, the Informed Deep Abstaining Classifier (IDAC) (Schneider et al., 2024a), this paper makes several contributions to advance noise-robust medical image segmentation:

- **Adaptation of Abstention to Segmentation**: We investigate the applicability of the abstention mechanism to image segmentation by adapting the DAC and IDAC loss functions for this domain.

- **Enhanced and Generalized Abstention Definition**: Our contribution improves and generalizes abstention, incorporating an informed regularization term guided by estimated noise rates $\tilde{\eta}$ and a power-law-based $\alpha$ auto-tuning algorithm.

- **Loss-Agnostic Integration and Novel Loss Functions**: We integrate the enhanced abstention mechanism with other loss functions, including Generalized Cross Entropy (GCE) (Zhang and Sabuncu, 2018), Symmetric Cross Entropy (SCE) (Wang et al., 2019), and Dice Loss (Milletari et al., 2016), introducing three loss functions: the Generalized Abstaining Classifier (GAC), the Symmetric Abstaining Classifier (SAC), and the Abstaining Dice Segmenter (ADS). ADS introduces architectural adaptations for class-wise abstention and class-specific noise rates $\tilde{\eta}_c$.

- **Empirical Validation of Robustness and Versatility**: Through empirical evaluations and quantitative analysis (Figure 1) on medical image datasets (CaDIS (Grammatikopoulou et al., 2021) and DSAD (Carstens et al., 2023)) under varying noise levels, we show consistent superiority over non-abstaining baselines.

The remainder of this paper reviews related work (Section 2), details our proposed abstention framework and novel loss functions (Section 3), outlines the experimental setup (Section 4), and presents a comprehensive evaluation of our results (Section 5) before concluding in Section 6.

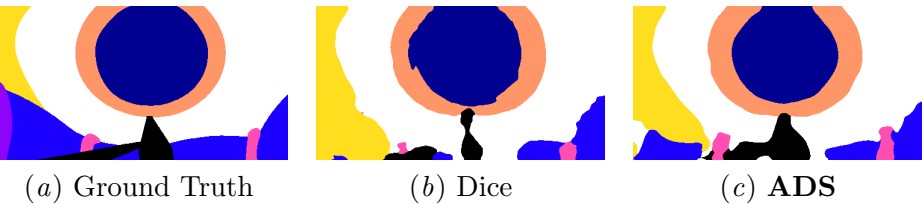

(a) Ground Truth          (b) Dice          (c) **ADS**

Figure 1: The impact of our noise-robust abstention framework. On a CaDIS sample with 25% label noise, the baseline Dice Loss (b) produces a noisy and inaccurate mask. In contrast, our proposed **Abstaining Dice Segmenter (ADS)** (c) yields a result that is visually cleaner and adheres more closely to the ground truth (a).

## 2. Related Work

The deployment of deep learning in medical imaging is frequently hampered by data imperfections, ranging from data scarcity and class imbalance (Tomar et al., 2025a,b) to missing or uncertain annotations (Schneider et al., 2023). Within this landscape, label noise remains a particularly pervasive challenge that has been explored through several avenues (Karimi et al., 2020).

Early strategies operated at different granularities, seeking to identify and correct noisy labels. These included pixel-wise adaptive weight maps, as proposed by Shi and Wu (2021), which dynamically adjust the contribution of each pixel to the loss, and graph-based label correction by Yi et al. (2022), which models spatial relationships to propagate corrections from reliable to unreliable pixels. Other frameworks operated at the image level to assess overall annotation quality, with some combining both pixel- and image-level perspectives to distill supervision more effectively (Shi and Wu, 2021; Zhu et al., 2019). Recognizing that segmentation noise is often not random but spatially correlated, other works have proposed explicit noise models. The Markov models from Yao et al. (2023), for instance, simulate realistic boundary distortions, while methods like LVC-Net by Shu et al. (2019) leverage local visual cues to guide the network away from incorrect labels during training.

A significant approach exploits the intrinsic learning dynamics of deep networks, particularly the 'early-learning' phenomenon, where models tend to fit clean, simple patterns before eventually memorizing the noise present in incorrect labels (Liu et al., 2022; Ye et al., 2024). This observation has led to the development of adaptive correction methods like ADELE by Liu et al. (2022), which detects the onset of memorization for each semantic class to intervene at the optimal moment. In a similar vein, multi-network and co-training paradigms leverage the consensus or disagreement between two or more models to filter out noisy signals. By using diverse architectures, these methods reduce the risk of confirmation bias, where a single model reinforces its own errors (Li et al., 2023; Rong et al., 2023).

Furthermore, emerging paradigms reframe the problem by treating large-scale noisy labels not as a hindrance but as a valuable resource. Pretraining strategies use massive, imperfectly labelled datasets to learn robust feature representations that can be fine-tuned on smaller, clean datasets (Liu et al., 2025). Other methods use meta-learning to bootstrap robust models, such as L2B by Zhou et al. (2024), which learns to dynamically weight the influence of observed labels and model-generated pseudo-labels during training.

An alternative and more fundamental approach that is most relevant to our work involves designing inherently robust loss functions. Instead of relying on external modules for noise detection or correction, this strategy embeds noise tolerance directly into the optimization objective (Karimi et al., 2020). Examples include the T-Loss from Gonzalez-Jimenez et al. (2023), which is based on the heavy-tailed Student-t distribution to reduce the influence of outliers, and the Active Negative Loss (ANL) framework proposed by Ye et al. (2024). Our work contributes to this line of research, but with a distinct and more modular philosophy. Instead of designing a new loss function from scratch, we propose a modular mechanism that can enhance the inherent robustness of existing losses.

A promising strategy for mitigating the impact of label noise is to empower a model to abstain from making a prediction on samples it deems unreliable. This approach circumvents the core problem of standard supervised learning, where a model is forced to commit to a prediction, potentially leading it to memorize erroneous labels. The concept was formally introduced for deep learning in the Deep Abstaining Classifier (DAC) by Thulasidasan et al. (2019). The DAC framework enables abstention by augmenting a network's architecture with an additional $(k+1)$-th output neuron, which explicitly represents the choice to abstain. Its corresponding loss function is defined as:

$$\mathcal{L}_{DAC}(x_j) = (1 - p_{k+1})\left(-\sum_{i=1}^{k} t_i \log \frac{p_i}{1 - p_{k+1}}\right) + \alpha \log \frac{1}{1 - p_{k+1}} \tag{1}$$

where $t_i$ is the ground truth label for class $i$. The loss is composed of two competing terms. The first is a modified Cross Entropy (CE) loss scaled by $(1 - p_{k+1})$, which represents the confidence in *not* abstaining, while the classification probability $p_i$ for each class $i$ is re-normalized by the same factor. The second component is a regularization term that directly penalizes the act of abstaining, where $p_{k+1}$ is the model's predicted probability of abstention (Thulasidasan et al., 2019).

The abstention penalty is controlled by the hyperparameter $\alpha$. Crucially, DAC employs an adaptive auto-tuning schedule where, after an initial warm-up period, $\alpha$ is initialized to a small value and is linearly increased over the remaining epochs up to a predefined value, $\alpha_{final}$. This ramp-up strategy, detailed in Appendix A, acts as a form of curriculum learning, initially permitting the model to ignore noisy samples and progressively forcing it to learn from more challenging data as its confidence grows (Thulasidasan et al., 2019).

Building directly upon this foundation, Schneider et al. (2024a) proposed the Informed Deep Abstaining Classifier (IDAC) to create a more targeted response to noise. IDAC refines the abstention mechanism by incorporating an a priori estimation of the dataset's noise level, $\tilde{\eta}$, directly into its regularization term. The IDAC loss functions is defined as:

$$\mathcal{L}_{IDAC}(x_j) = (1 - p_{k+1})\left(-\sum_{i=1}^{k} t_i \log \frac{p_i}{1 - p_{k+1}}\right) + \alpha(\tilde{\eta} - \hat{\eta})^2 \tag{2}$$

The key innovation lies in replacing DAC's incremental penalty with a term that minimizes the divergence between the expected noise rate $\tilde{\eta}$ and the model's current batch-wise abstention rate, $\hat{\eta}$. This provides a more direct supervisory signal, guiding the model to abstain on a fraction of samples that is consistent with the known level of label corruption (Schneider et al., 2024a).

While DAC and IDAC have demonstrated the profound effectiveness of abstention, their application has been confined to the CE loss paradigm. Our work addresses this limitation by proposing a generalized abstention framework, establishing it as a modular tool to enhance the robustness of a diverse range of loss functions.

## 3. The Universal Abstention Framework

Building upon the demonstrated efficacy of DAC and IDAC in mitigating label noise through abstention, we propose an enhanced and universal definition of the abstention mechanism that can be readily adapted to virtually any underlying loss function, $\mathcal{L}_X(x_j)$. Our generalized abstaining loss is formulated as:

$$\mathcal{L}_{abstention}(x_j) = (1 - p_{k+1})\mathcal{L}_X(x_j) + \alpha \left| \log \frac{1 - \tilde{\eta}}{1 - p_{k+1}} \right| \tag{3}$$

With this formulation, we introduce two critical innovations designed to provide greater flexibility and more targeted noise mitigation.

### 3.1. Informed Regularization

The first improvement lies in the **regularization term** $\alpha \left| \log \frac{1-\tilde{\eta}}{1-p_{k+1}} \right|$. This term draws inspiration from IDAC by explicitly incorporating the expected noise rate $\tilde{\eta}$ to guide the abstention behaviour. Unlike DAC, which pushes the abstention probability $p_{k+1}$ toward zero, our term incentivises the model to maintain $p_{k+1}$ in proximity to $\tilde{\eta}$. This allows the model to continue abstaining on samples it confidently perceives as noisy, rather than being forced to make classification decisions that could elevate the risk of overfitting to noise. This enhanced definition of the regularization term is also flexible; if a reliable estimate for $\tilde{\eta}$ isn't available, setting $\tilde{\eta} = 0$ effectively reduces the term to its original DAC form, which has already demonstrated its strength and effectiveness in combating label noise.

### 3.2. Power-Law Auto-Tuning

The second and more significant enhancement concerns the **$\alpha$ auto-tuning algorithm**. The original algorithm proposed by DAC employed a linear ramp-up strategy for $\alpha$ after a warm-up phase, which, while effective, offered limited flexibility in controlling the learning trajectory. Our refined approach replaces this with a simpler yet more powerful and flexible method. For every epoch $e$ after an initial warm-up phase of $L$ epochs out of a total $E$ epochs, $\alpha$ is dynamically calculated as:

$$\alpha = \alpha_{final} * \left( \frac{e - L}{E - L} \right)^{\gamma} \tag{4}$$

In this equation, $\gamma > 0$ serves as a growth factor that precisely controls the rate at which $\alpha$ increases throughout the abstention phase, as depicted in Appendix B.1. The behaviour of $\alpha$ is modulated by $\gamma$: if $\gamma > 1$, $\alpha$ exhibits a sublinear growth, increasing slowly at the beginning of the abstention period and accelerating its growth towards the end of training. This behaviour intensifies with larger values of $\gamma$. Conversely, if $\gamma < 1$, $\alpha$ experiences superlinear growth early in the abstention phase, with its rate of increase slowing down as training progresses. Setting $\gamma = 1$ yields a linear increment, akin to DAC's approach. This formulation provides significant flexibility in penalizing and guiding the abstention behaviour, enabling a more optimal balance between the model's learning from clean data and its strategic abstention from noisy or ambiguous samples.

### 3.3. Novel Abstaining Loss Functions for Segmentation

We demonstrate our framework's versatility by creating three novel, noise-robust loss functions, one of which is tailored for segmentation.

#### 3.3.1. Abstaining Classifiers (GAC and SAC)

We first integrate our framework with two CE-based losses. The **G**eneralized **A**bstaining **C**lassifier (GAC) combines abstention with Generalized Cross Entropy (GCE) (Zhang and Sabuncu, 2018), creating a dual defence where GCE's bounded loss attenuates noise on classified samples, while abstention filters out the most corrupted ones. The **S**ymmetric **A**bstaining **C**lassifier (SAC) enhances Symmetric Cross Entropy (SCE) (Wang et al., 2019), empowering the model to completely disengage from highly suspect samples, rather than merely re-balancing their influence. SAC can actively filter out the most egregious noisy examples, allowing the symmetrical CE-RCE components to focus on refining predictions for the more reliable data.

#### 3.3.2. Abstaining Dice Segmenter (ADS)

Our most significant adaptation is the **A**bstaining **D**ice **S**egmenter (ADS)[1], which integrates our framework with the region-based Dice loss (Milletari et al., 2016). This required two fundamental architectural changes to resolve the incompatibility between Dice's class-wise nature and standard pixel-wise abstention:

- **Class-wise Abstention Head:** We re-conceptualized the network's output to produce class-wise abstention predictions. As illustrated in Figure 2(b), a specialized module uses Adaptive Average Pooling with an output size $s \times s$, followed by a Linear layer and *sigmoid* activation to output a unique abstention probability for each of the $k$ classes.

- **Class-specific Regularization:** To complement the class-wise abstention, the regularization term in Equation (3) is formulated to accept a vector of class-specific noise estimates, $\tilde{\eta}_c$ (calculation detailed in Appendix D.1). This enables granular control over abstention behavior per anatomical structure. Crucially, the design preserves

---

1. The *Segmenter* in ADS highlights its design for segmentation tasks, contrasting with the other 'Classifier' losses which can also be used for classification.

flexibility: if detailed class-wise statistics are infeasible to obtain, ADS seamlessly accepts a single global noise estimate $\tilde{\eta}$ applied uniformly across all classes. However, as we will demonstrate in Section 5.1 and Appendix D.1, this scalar simplification can be suboptimal given the high inter-class noise variance typical in segmentation. In such scenarios, a superior strategy would be to employ methods that dynamically estimate noise rates from the data, such as Confident Learning (Northcutt et al., 2021), Beta Mixture Models (Arazo et al., 2019), or Transition Matrix Estimation (Xia et al., 2019).

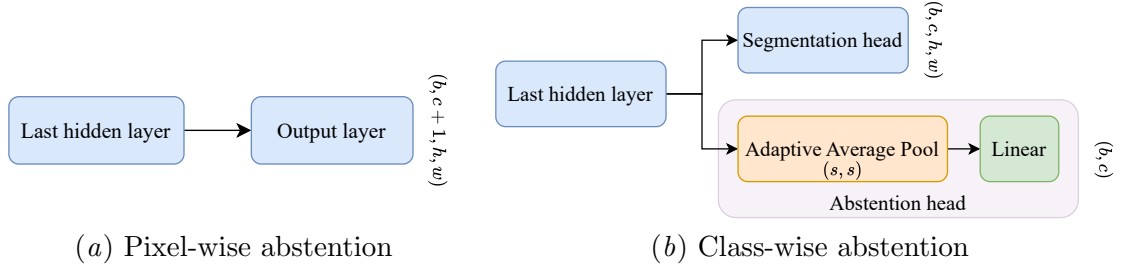

$(a)$ Pixel-wise abstention $\qquad\qquad$ $(b)$ Class-wise abstention

Figure 2: Transforming the output layer from standard pixel-wise abstention (a) to our proposed class-wise abstention head for **ADS** (b). The dimensions $b, c, h, w$ represent batch size, number of classes or channels, height, and width, respectively. $(s, s)$ is the output size for the Adaptive Average Pool layer.

## 4. Experimental Setup

We validated our framework on two publicly available surgical datasets with distinct characteristics to demonstrate the generalizability of our approach.

- Cataract Dataset for Image Segmentation (CaDIS): A benchmark featuring 4,670 frames from cataract surgery with dense, high-quality pixel-wise annotations (Grammatikopoulou et al., 2021). For our experiments, we utilized the 8-class variant, which groups surgical tools into a single class. Images were normalized and resized to $480 \times 256$.

- Dresden Surgical Anatomy Dataset (DSAD): A more complex benchmark with 1,430 frames from laparoscopic surgery (Carstens et al., 2023). This dataset presents a greater challenge due to its intricate anatomical structures, sparse annotations (approx. 82% background), and significant class imbalance. Images were normalized and resized to $480 \times 384$.

To rigorously assess noise robustness, we simulated realistic annotation errors with a two-pronged approach. Structural noise was introduced via morphological transformations (erosion and dilation) to simulate boundary inaccuracies, while semantic noise was injected via stochastic label flipping to mimic annotator bias (Karimi et al., 2020; Zhang et al., 2020;

Marcinkiewicz and Mrukwa, 2019; Li et al., 2023). Visualizations demonstrating the realism and severity of these structural and semantic corruptions, including difference maps between ground truth and noisy annotations, are provided in Appendix D.

We evaluated performance across five calibrated noise levels for each dataset: 5-25% corruption for CaDIS and 3-15% for DSAD. Specifically, we define the noise rate $\eta$ as the global percentage of pixels in the dataset where the noisy annotation mask differs from the ground truth mask (i.e., $\frac{1}{N} \sum \mathbb{1}(\hat{y} \neq y) \approx \eta$).

We used an NVIDIA A100 80GB to train a U-Net model (Ronneberger et al., 2015) with a pretrained ResNet-50 (He et al., 2016) backbone for our experiments. Key training hyperparameters are detailed in Appendix C.1. To ensure statistical reliability, each experiment was conducted five times with distinct random seeds. Hyperparameters for each loss function were optimized to yield the highest validation mean Intersection over Union (mIoU) on the highest noise level for each dataset, thereby maximizing noise resistance. Crucially, to ensure a fair comparison and rigorously isolate the impact of our abstention framework, the optimal hyperparameters found for the baseline GCE and SCE functions were deliberately held constant for their respective abstaining versions, GAC and SAC. While likely suboptimal for our novel functions, this methodology ensures that any observed performance gains are attributable solely to the abstention mechanism itself. Appendix C.2 details the hyperparameters we used for each loss in our benchmarks.

## 5. Evaluations and Visual Analysis

The efficacy of our proposed framework was validated through extensive experiments on the CaDIS and DSAD datasets. The results, summarized in Table 1 and Figure 3, demonstrate the superior noise-robustness conferred by the abstention mechanism. As anticipated, all loss functions exhibited a performance decline with increasing label noise, underscoring the universal challenge of learning from corrupted data. However, the critical distinction lies in the rate of this degradation. Our proposed abstaining loss functions consistently demonstrated a more graceful performance decline and maintained a significant advantage over their respective non-abstaining baselines, particularly at high noise intensities. On the CaDIS dataset at 25% noise, our Abstaining Dice Segmenter (ADS) emerged as the top performer, achieving a 5.35% mIoU lead over the standard Dice Loss. Similarly, GAC and SAC surpassed their baselines, confirming the broad applicability of our framework. This trend persisted on the more complex DSAD dataset, where despite lower overall mIoU scores, the abstaining variants maintained a clear and consistent performance advantage, highlighting their robust effectiveness even in challenging segmentation scenarios.

The flatter degradation curves for the abstaining methods in Figure 3 highlight their superior resilience. This observation is quantitatively substantiated in Table 2, where we report the normalized performance drop rate ($\Delta mIoU/\Delta\eta$). Most notably, ADS demonstrated a statistically significant reduction in degradation compared to the Dice baseline on both datasets (CaDIS: $p < 0.001$, DSAD: $p = 0.003$). The CE-based variants showed dataset-dependent improvements: SAC yielded significant gains on CaDIS ($p = 0.001$), while GAC proved significantly more robust on the challenging DSAD benchmark ($p = 0.009$). This confirms that while the framework is effective, the optimal choice of base loss may depend on dataset characteristics.

Table 1: Average test mIoU (%) and standard deviation across 5 runs of a U-Net model trained on CaDIS and DSAD datasets. We used the scalar noise rate $\tilde{\eta}(\approx \eta)$ for IDAC, GAC, and SAC, and the class-wise noise vector $\tilde{\eta}_c$ for ADS. **Gray background:** Abstaining loss functions. **(*):** Our proposed novel loss functions. **Structure:** The table is divided into four comparative groups (separated by double vertical lines); each group compares a baseline loss against its abstaining counterpart(s). **Bold:** Indicates the best result *within that specific group*. For example, in the last group, we compare Dice vs. ADS to isolate the impact of our framework on the Dice loss.

| Dataset | Noise rate $\eta$ (%) | Loss function | | | | | | | | |
|---|---|---|---|---|---|---|---|---|---|---|
| | | CE | DAC | IDAC | GCE | GAC* | SCE | SAC* | Dice | ADS* |
| CaDIS | 0 | **76.02±0.70** | 75.29±0.79 | 75.36±0.73 | 73.49±3.27 | **73.76±2.80** | 75.38±0.75 | **75.83±0.62** | 76.52±0.47 | **77.04±0.37** |
| | 5 | **73.67±1.03** | 73.14±0.46 | 72.89±0.41 | **72.83±1.11** | 71.73±2.79 | 73.41±0.71 | **73.51±1.59** | 73.48±0.28 | **75.22±0.85** |
| | 10 | 66.39±0.17 | **67.43±0.49** | 66.92±0.49 | **64.82±0.86** | 64.16±2.57 | 65.92±0.91 | **67.29±1.65** | 66.51±0.61 | **71.12±0.55** |
| | 15 | 64.15±2.47 | **65.85±1.05** | 64.87±0.91 | **64.81±0.46** | 64.44±2.70 | 62.16±1.99 | **65.48±2.11** | 67.31±0.73 | **70.80±1.08** |
| | 20 | 59.56±1.21 | **63.42±0.87** | 60.54±2.27 | 60.73±1.41 | **60.91±1.64** | 57.62±4.22 | **62.70±0.31** | 63.64±0.82 | **68.88±0.49** |
| | 25 | 52.27±1.70 | **60.63±2.73** | 58.19±4.77 | 55.71±1.30 | **59.46±0.76** | 55.08±0.93 | **61.27±1.22** | 61.04±1.41 | **66.39±0.67** |
| DSAD | 0 | **34.25±2.50** | 34.01±0.96 | 33.60±0.72 | **35.14±1.65** | 32.26±0.53 | 32.78±1.19 | **33.86±1.83** | 31.28±0.87 | 30.09±1.10 |
| | 3 | **33.69±1.85** | 33.67±2.01 | 32.76±2.03 | **33.84±2.56** | 32.94±2.23 | **32.11±1.09** | 30.90±2.76 | **30.83±4.78** | 28.64±2.76 |
| | 6 | **30.70±2.47** | 29.47±1.97 | 29.11±2.10 | 29.69±1.96 | **29.78±4.27** | 30.51±2.16 | **31.55±2.43** | 28.56±1.00 | **30.48±3.61** |
| | 9 | **24.65±2.90** | 24.58±2.61 | 23.47±2.48 | 22.95±2.93 | **28.84±4.17** | 28.02±2.37 | **28.55±1.29** | 19.04±1.92 | **26.23±2.05** |
| | 12 | 21.00±3.15 | **22.59±4.35** | 20.94±1.86 | 19.84±2.89 | **25.00±4.13** | 21.57±0.67 | **23.73±0.68** | 16.15±1.49 | **22.63±0.51** |
| | 15 | 14.41±2.59 | **17.69±3.97** | 16.24±1.45 | 14.12±2.91 | **20.01±2.56** | 15.31±0.75 | **15.91±3.53** | 14.65±1.50 | **18.05±1.63** |

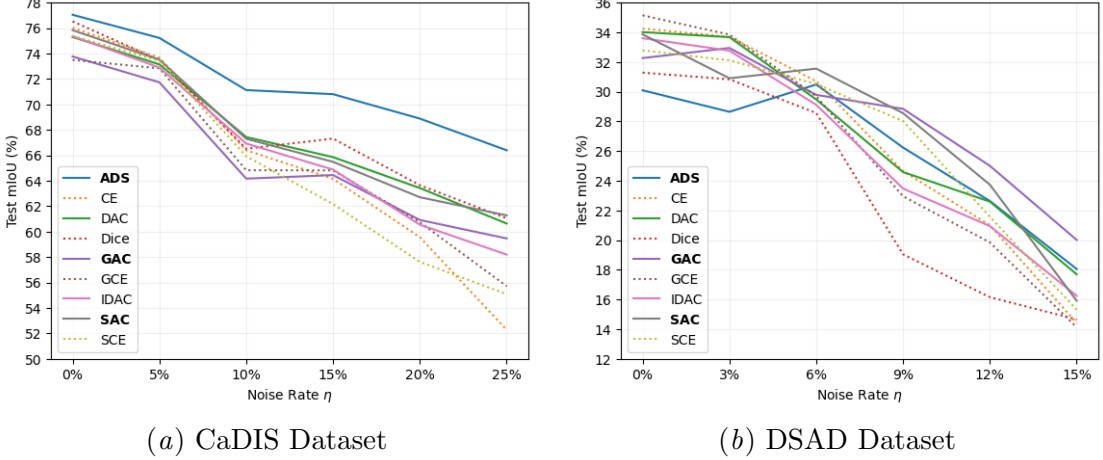

(*a*) CaDIS Dataset          (*b*) DSAD Dataset

Figure 3: Quantitative comparison of noise-robustness. The plots show the average test mIoU (%) degradation as label noise $\eta$ increases. The flatter curves of the abstaining variants (solid lines) demonstrate their superior resilience compared to non-abstaining baselines (dashed lines). Our proposed losses (GAC, SAC, ADS) are in **bold**.

A qualitative review of the segmentation masks, depicted in Figures 4 and 5 further substantiates these quantitative gains. Visual inspection of the CaDIS results revealed that models trained with our abstaining losses produced markedly cleaner and more accurate segmentations than their baselines. Contours were sharper, noise artifacts were reduced, and overall structural coherence was improved. Most notably, ADS produced masks with the highest to similarity the ground truth even at the highest noise level. While the inherent

difficulty of the DSAD dataset resulted in lower-quality predictions across all methods, the same relative improvements were observed. The abstaining models consistently generated more coherent masks with fewer spurious predictions and better-defined boundaries compared to their non-abstaining counterparts. This visual evidence confirms that the improvements in mIoU translate directly to more reliable and clinically relevant segmentation outputs.

Table 2: Quantitative analysis of robustness using the **Normalized Performance Drop Rate** ($\Delta$mIoU$/\Delta\eta$) across 5 runs on CaDIS and DSAD. Values represent the average mIoU points lost for every 1% increase in label noise (Mean$\pm$95% CI over 5 seeds). Lower values indicate greater resilience. **Gray background:** Abstaining loss functions. **(*):** Our proposed methods. **Structure:** The table is grouped to compare baseline losses against their abstaining counterparts. **(†):** Statistically significant improvement over baseline ($p < 0.05$, paired t-test).

| Loss Function | CaDIS | DSAD |
|---|---|---|
| CE | $0.950\pm0.099$ | $1.323\pm0.379$ |
| DAC | $0.587\pm0.167^{\dagger}$ | $1.088\pm0.346$ |
| IDAC | $0.687\pm0.255$ | $1.157\pm0.149$ |
| GCE | $0.711\pm0.140$ | $1.401\pm0.166$ |
| GAC* | $0.572\pm0.140$ | $0.817\pm0.197^{\dagger}$ |
| SCE | $0.812\pm0.068$ | $1.165\pm0.075$ |
| SAC* | $0.582\pm0.046^{\dagger}$ | $1.197\pm0.202$ |
| Dice | $0.619\pm0.079$ | $1.108\pm0.154$ |
| ADS* | $0.426\pm0.036^{\dagger}$ | $0.803\pm0.082^{\dagger}$ |

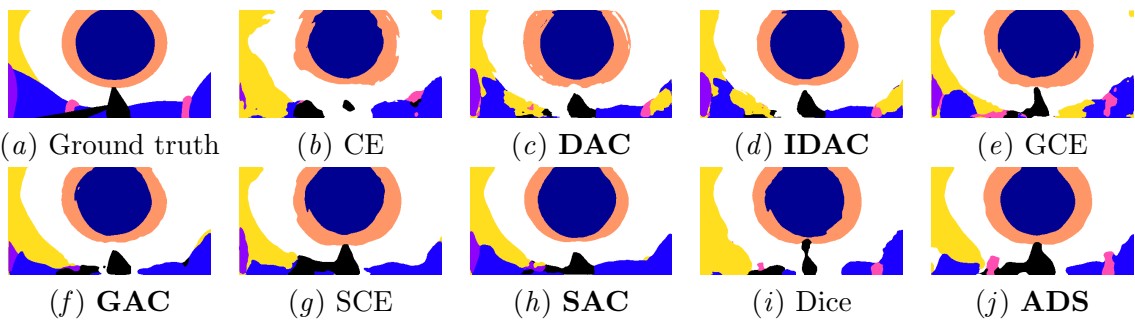

$(a)$ Ground truth  $(b)$ CE  $(c)$ **DAC**  $(d)$ **IDAC**  $(e)$ GCE

$(f)$ **GAC**  $(g)$ SCE  $(h)$ **SAC**  $(i)$ Dice  $(j)$ **ADS**

Figure 4: Qualitative comparison on a CaDIS sample at 25% noise. Our proposed abstaining losses (**GAC**, **SAC**, **ADS**) produce masks with higher fidelity and fewer artifacts than their baselines. Abstaining losses are in **bold**.

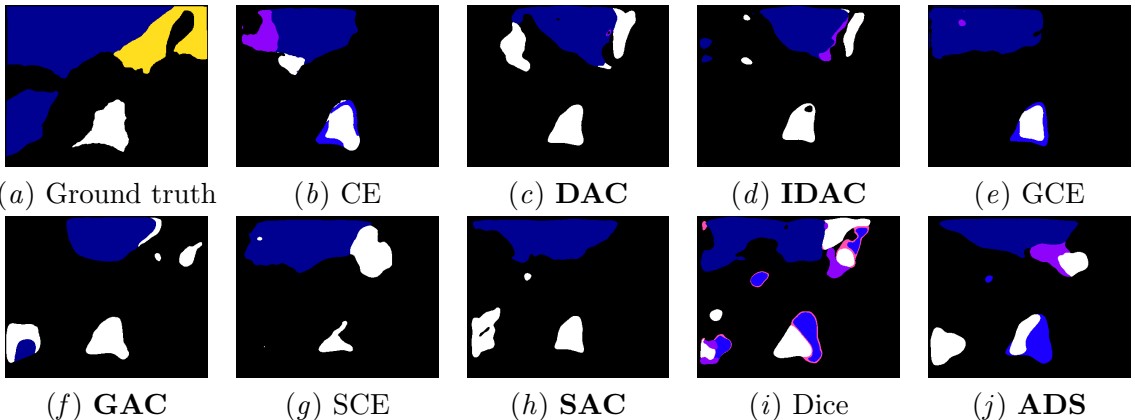

Figure 5: Qualitative comparison on a challenging DSAD sample at 15% noise. Abstaining variants (in **bold**) yield masks with better structural coherence and fewer spurious activations than their baselines.

### 5.1. Sensitivity to Prior Noise Rate Estimation

A practical concern for deployment is the potential unavailability of an accurate noise rate prior, $\tilde{\eta}$. To evaluate the sensitivity of our framework to this hyperparameter, we conducted an ablation study on both CaDIS (at true $\eta = 25\%$) and DSAD (at true $\eta = 15\%$), where we deliberately mis-specified the estimated $\tilde{\eta}$. For this analysis, all three losses (GAC, SAC, and ADS) utilized a single **scalar** global noise estimate to ensure a consistent comparison.

Table 3 presents the results, comparing the "Oracle" setting ($\tilde{\eta} \approx \eta$) against under-estimation and over-estimation. SAC displayed remarkable stability, with performance remaining virtually unchanged across all priors. GAC and ADS showed a dependency on the prior, yet even in the extreme case of an uninformed prior ($\tilde{\eta} = 0$, effectively reverting to a DAC-style regularizer with our power-law schedule), both methods outperformed their non-abstaining baselines.

It is notable that the peak ADS performance in this scalar sweep (63.07% on CaDIS) is lower than the main result reported in Table 1 (66.39%), which used class-specific noise vectors $\tilde{\boldsymbol{\eta}_c}$. This discrepancy highlights the importance of the class-wise formulation. As shown in Appendix D.1, both datasets contain extreme inter-class noise variance (ranging from 9.7% to 91.1% for CaDIS, and from 5.7% to 94.6% for DSAD) at their highest noise levels. Using a single scalar prior (e.g., 25%) inevitably under-estimates the noise for difficult classes, hampering the model's ability to abstain effectively on them. However, the results confirm that even without this granular information, ADS remains effective and robust to the scalar estimate itself.

Table 3: Sensitivity analysis of GAC, SAC, and ADS under varying scalar estimated noise rates $\tilde{\eta}$. The "Oracle" settings are underlined. While SAC is nearly invariant to $\tilde{\eta}$, GAC and ADS show moderate sensitivity but remain robust even when $\tilde{\eta} = 0$. Note that for this experiment we employed the global scalar noise rate $\tilde{\eta}$ for ADS, unlike Table 1 where we used $\tilde{\eta}_c$ to showcase ADS's capabilities under optimal conditions.

| Dataset | $\tilde{\eta}$ (%) | Loss function | | |
| --- | --- | --- | --- | --- |
| | | GAC | SAC | ADS |
| CaDIS | 0 | 57.34±1.12 | 61.27±1.23 | 61.53±1.41 |
| | 15 | 59.35±0.78 | 61.27±1.22 | 62.55±1.00 |
| | 25 (Oracle) | 59.46±0.76 | 61.27±1.22 | 63.07±0.96 |
| | 35 | 59.47±0.73 | 61.27±1.22 | 63.51±0.92 |
| | 50 | 59.77±0.99 | 61.28±1.22 | 63.90±1.12 |
| DSAD | 0 | 20.30±2.86 | 15.87±3.46 | 18.29±2.14 |
| | 10 | 20.11±2.66 | 15.87±3.47 | 17.94±1.95 |
| | 15 (Oracle) | 20.01±2.56 | 15.91±3.53 | 18.10±2.14 |
| | 20 | 20.18±2.71 | 15.81±3.39 | 17.71±1.55 |
| | 30 | 19.02±3.22 | 15.91±3.54 | 18.03±1.45 |

## 6. Conclusions

In this paper, we have established that the abstention mechanism is not merely a classification trick, but a fundamental and robust computer vision strategy whose capabilities are exceptionally beneficial for medical image segmentation. By introducing a universal framework equipped with informed regularization and dynamic auto-tuning, we have successfully generalized abstention to function with diverse loss paradigms, including the region-based Dice loss. This modularity allows for the creation of specialized noise-robust loss functions without requiring complex, computationally expensive changes to the underlying model architecture.

Our extensive validation on the CaDIS and DSAD datasets confirms that this approach yields consistent and significant improvements over standard baselines, particularly in high-noise situations where traditional models fail. The ability of the ADS model to produce clean, anatomically coherent masks even when trained on 25% corrupted labels highlights the practical value of allowing models to selectively ignore unreliable supervision.

While our results are promising, we acknowledge that our evaluation relied on synthetic noise injection, which serves as a controlled proxy for the complex, structured ambiguity often found in clinical annotations. Furthermore, the current framework relies on a pre-estimated noise rate hyperparameter. Future work will focus on validating this framework on datasets with naturally occurring inter-rater variability and developing adaptive, data-driven methods to estimate the noise rate dynamically. Ultimately, this work provides a scalable pathway toward building trustworthy diagnostic systems that can robustly learn from the imperfect data realities of the medical domain.

## Acknowledgments

This research has been funded by the Federal Ministry of Education and Research of Germany and the state of North-Rhine Westphalia as part of the Lamarr-Institute for Machine Learning and Artificial Intelligence.

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

## Appendix A. DAC $\alpha$ auto-tuning algorithm

For reference and comparison, Algorithm 1 outlines the original linear auto-tuning schedule proposed by Thulasidasan et al. (2019). As illustrated, this approach requires a stateful, iterative update process where $\alpha$ is initialized based on performance during a warm-up phase (Lines 3-9) and then incremented by a fixed $\delta_\alpha$ at each step (Lines 16-19). In contrast, our proposed framework (Section 3) simplifies this process significantly. By replacing this iterative logic with the direct power-law formulation in Equation (4), we eliminate the need for state tracking and intermediate variable initialization, while simultaneously offering greater flexibility in the curriculum schedule.

---

**Algorithm 1:** $\alpha$ auto-tuning

---

**Input:** total iter. $(T)$, current iter. $(t)$, total epochs $(E)$, abstention-free epochs $(L)$, current epoch $(e)$, $\alpha$ init factor $(\rho)$, final $\alpha$ $(\alpha_{final})$, mini-batch cross-entropy over true classes $(\mathcal{H}_c(P^M_{1...K}))$

**1** $\alpha_{set} = $ False;
**2 for** $t := 0$ **to** $T$ **do**
**3** $\quad$ **if** $e < L$ **then**
**4** $\quad\quad$ $\beta = (1 - P^M_{k+1})\mathcal{H}_c(P^M_{1...K})$;
**5** $\quad\quad$ **if** $t = 0$ **then**
**6** $\quad\quad\quad$ $\tilde{\beta} = \beta$ ; $\qquad\qquad\qquad$ // {initialize moving average }
**7** $\quad\quad$ **end**
**8** $\quad\quad$ $\tilde{\beta} \leftarrow (1 - \mu)\tilde{\beta} + \mu\beta$;
**9** $\quad$ **end**
**10** $\quad$ **if** $e = L$ **and not** $\alpha_{set}$ **then**
**11** $\quad\quad$ $\alpha := \tilde{\beta}/\rho$ ; $\qquad\qquad$ // {initialize $\alpha$ at start of epoch $L$ }
**12** $\quad\quad$ $\delta_\alpha := \frac{\alpha_{final} - \alpha}{E - L}$;
**13** $\quad\quad$ $update_{epoch} = L$;
**14** $\quad\quad$ $\alpha_{set} = $ True;
**15** $\quad$ **end**
**16** $\quad$ **if** $e > update_{epoch}$ **then**
**17** $\quad\quad$ $\alpha \leftarrow \alpha + \delta_\alpha$ ; $\qquad\qquad$ // {then update $\alpha$ once every epoch }
**18** $\quad\quad$ $update_{epoch} = e$;
**19** $\quad$ **end**
**20 end**

---

## Appendix B. Abstention Dynamics during Training

To better understand how different loss functions utilize the abstention mechanism throughout the training process, we visualized the batch-wise abstention rate over time. Figure 6 depicts the training trajectory for DAC, IDAC, and our proposed GAC on the CaDIS dataset with a synthetic noise rate of $\eta = 15\%$. The plot reveals distinct behaviours after the warm-up phase. The original DAC (blue) exhibits a rapid collapse in abstention after an initial spike. The penalty forces the abstention rate effectively to zero, meaning the model stops utilizing the mechanism and risks overfitting to noisy labels. IDAC (orange) avoids zero, but exhibits high variance. In contrast, our proposed GAC (green) demonstrates a controlled and graceful descent, eventually stabilizing at an abstention rate of approximately 15%.

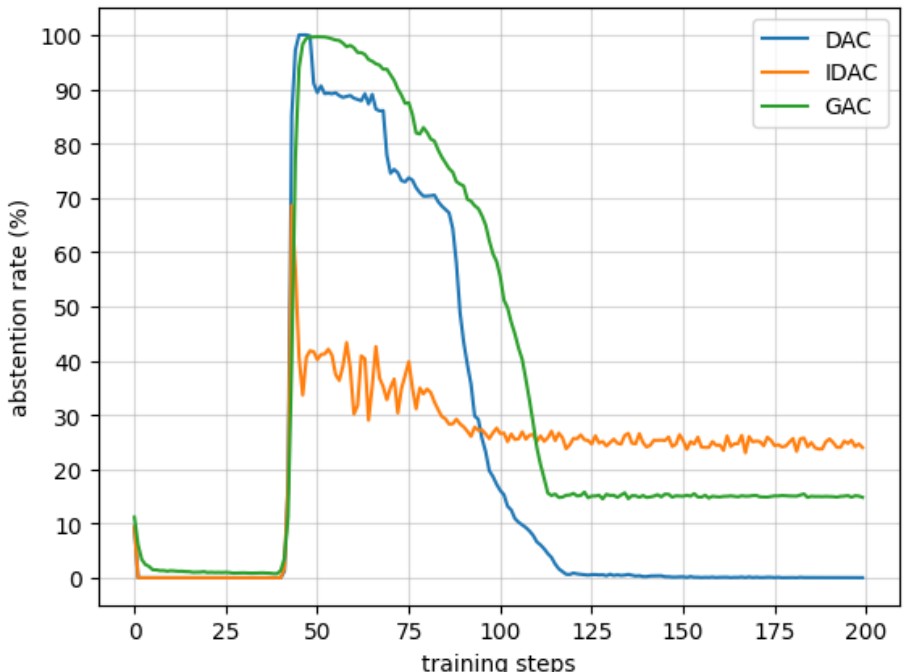

Figure 6: Evolution of the abstention rate during training on the CaDIS dataset with 15% label noise.

### B.1. Alpha Auto-Tuning Behaviour

As described in Section 3.2, our framework utilizes a power-law-based auto-tuning algorithm for the abstention penalty $\alpha$. Figure 7 visually demonstrates the effect of the growth factor $\gamma$ on the trajectory of $\alpha$ throughout the training process, enabling sublinear, linear, or superlinear growth.

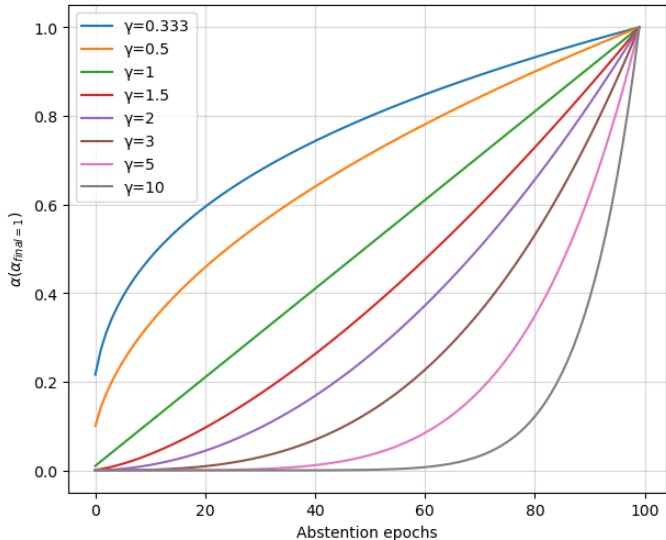

Figure 7: The effect of different values of $\gamma$ on the growth of $\alpha$ with $\alpha_{final} = 1$.

## Appendix C. Additional Experimental Details and Method Parameters

This appendix provides supplementary details regarding our experimental setup and the parameters of our proposed abstention framework.

### C.1. Global Training Hyperparameters

All experiments in the main paper were conducted using a consistent set of global training parameters to ensure a fair comparison. These parameters, including the network architecture, optimizer, and learning rate schedule, are detailed in Table 4.

Table 4: Training hyperparameter configurations used in our experiments.

| Parameter | Value |
|---|---|
| Architecture | U-Net |
| Backbone | Pretrained ResNet-50 |
| Optimizer | AdamW |
| Epochs | 50 |
| Initial Learning Rate | 0.003 |
| LR Schedule | Step decay; factor of 0.2 every 10 epochs |
| Batch Size (CaDIS) | 128 |
| Batch Size (DSAD) | 50 |
| Seed Runs | 5 |

### C.2. Loss Function-Specific Hyperparameters

The hyperparameters for each loss function were selected through a rigorous two-stage optimization process on the validation set, using the highest noise level for each dataset ($\eta = 25\%$ for CaDIS, $\eta = 15\%$ for DSAD) to prioritize robustness. First, we conducted a Bayesian search using Weights & Biases to identify the high-performing ranges for each parameter. Second, based on these ranges, we performed a fine-grained grid search over a set of interpretable, discrete values (e.g., stepping $\alpha_{final}$ by 0.5 or 1.0) to select the final optimal configuration. This approach ensures that the chosen hyperparameters are both effective and generalizable, avoiding overfitting to specific float values found during random search. The final parameters used to generate the results in our paper are listed in Table 5.

Table 5: The hyperparameter configurations for each loss function. $L$ is the number of warm-up epochs, $\alpha$ is IDAC's fixed abstention penalty, and $\alpha_{final}$ is the target penalty for DAC, GAC, SAC, and ADS. $\gamma$ is the growth factor for our enhanced $\alpha$ auto-tuning algorithm, and $s$ is the pooling output size for the class-wise abstention module in ADS. Note that $\alpha$ represents the fixed abstention penalty for IDAC, and the Cross Entropy coefficient for SCE.

| Dataset | DAC | IDAC | GCE | GAC | SCE | SAC | ADS |
|---------|-----|------|-----|-----|-----|-----|-----|
| CaDIS | $\alpha_{final} = 1$
$L = 10$ | $\alpha = 1$
$L = 10$ | $q=0.5$ | $\alpha_{final} = 3$
$L = 10$
$\gamma = 3$ | $\alpha = 1$
$\beta = 1$ | $\alpha_{final} = 1$
$L = 10$
$\gamma = 1.5$ | $\alpha_{final} = 1$
$L = 10$
$\gamma = 3$
$s = 16$ |
| DSAD | $\alpha_{final} = 2$
$L = 18$ | $\alpha = 1$
$L = 10$ | $q=0.1$ | $\alpha_{final} = 2$
$L = 15$
$\gamma = 2$ | $\alpha = 0.5$
$\beta = 1$ | $\alpha_{final} = 1$
$L = 20$
$\gamma = 3$ | $\alpha_{final} = 4$
$L = 10$
$\gamma = 1.5$
$s = 16$ |

## Appendix D. Visualization of Synthetic Noise

To provide a visual assessment of the difficulty and realism of our synthetic noise injection protocol, we present qualitative examples from both the CaDIS and DSAD datasets in Figure 8 and Figure 9, respectively. The visualizations display the progression of corruption alongside **Difference Maps**, where white pixels indicate disagreement between the ground truth and the noisy mask. These maps clearly highlight that our noise generation strategy creates not only random semantic errors (large flipped regions) but also challenging structural artifacts along object boundaries, mimicking the inter-rater variability often seen in clinical annotations.

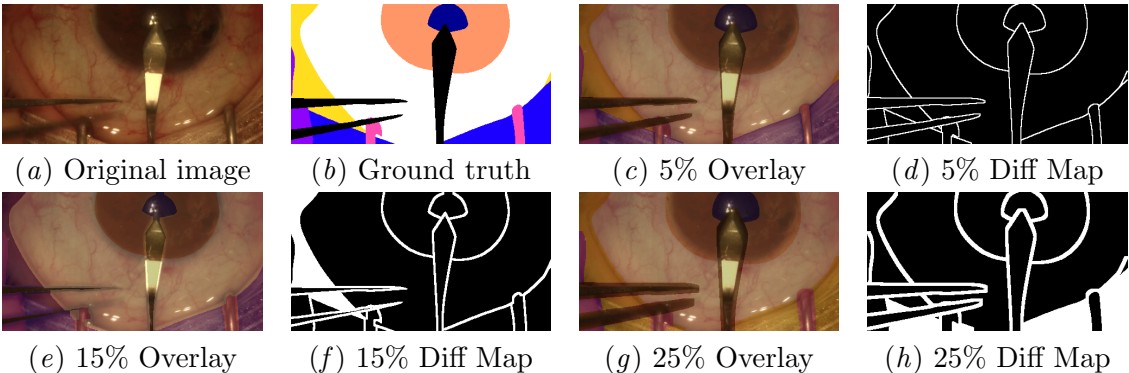

Figure 8: Visualization of synthetic label noise on a sample CaDIS frame. The figure illustrates the progression of corruption from low ($\eta = 5\%$) to severe ($\eta = 25\%$) noise levels.(a)-(b) Show the original surgical view and the clean ground truth. (c)-(h) Display the **Noisy Overlays** (transparent mask on image) and corresponding **Difference Maps** at 5%, 15%, and 25% noise. In the Difference Maps, **Black** indicates agreement with the ground truth, while **White** indicates corrupted pixels.

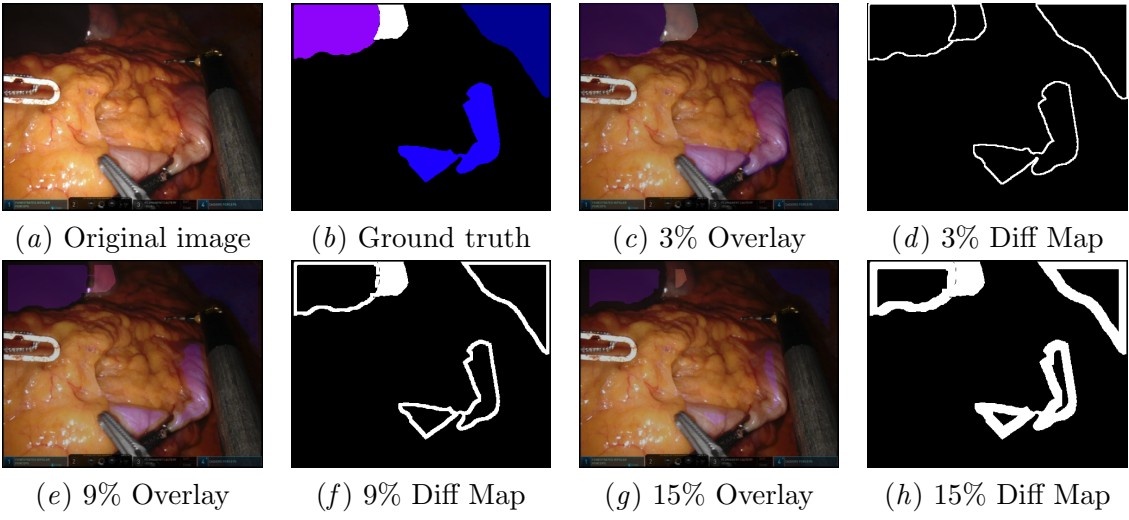

Figure 9: Visualization of synthetic label noise on a sample DSAD frame. This dataset presents a challenging scenario with sparse annotations and complex anatomy. (a)-(b) Show the raw laparoscopic image and the ground truth. (c)-(h) Illustrate the impact of noise at 3%, 9%, and 15% levels. The **Difference Maps** (White = Error) highlight that even at lower noise rates ($\eta = 3\%$), significant structural corruption is introduced along the organ boundaries. At higher rates ($\eta = 15\%$), the corruption creates large, misleading semantic regions that severely test model robustness.

### D.1. Class-wise Noise Distribution

To better understand the performance gap between the scalar-prior ADS (Table 3) and the vector-prior ADS (Table 1), we provide the calculated class-wise noise rates used for the vector experiments.

- CaDIS (at global $\eta = 25\%$): $[68.2\%, 19.6\%, 36.6\%, 80.5\%, 91.1\%, 41.4\%, 23.7\%, 9.7\%]$.

- DSAD (at global $\eta = 15\%$): $[5.7\%, 33.7\%, 66.7\%, 59.5\%, 94.6\%, 90.9\%, 92.1\%, 45.5\%]$.

The noise rate for a specific class $c$ as the fraction of pixels belonging to class $c$ in the ground truth that are corrupted in the noisy mask:

$$\tilde{\eta}_c = \frac{\sum_i \mathbb{1}(\hat{y}_i \neq c \wedge y_i = c)}{\sum_i \mathbb{1}(y_i = c)}$$

The high variance in these values demonstrates why a single scalar prior (e.g., 25%) can be suboptimal for ADS, as it drastically underestimates the corruption for certain classes (e.g., 91.1%), limiting the model's ability to abstain where it is needed most.

