# OpenReview forum: "Generalizing Abstention for Noise-Robust Learning in Medical Image Segmentation"
_MIDL.io/2026/Conference — MIDL 2026 Poster_

### Official Review · Reviewer_kVoa · 2026-01-10

**Confidence:** 3
**Preliminary Rating:** 3
**Final Rating:** 4

**Summary:**

This paper proposes a universal abstention framework (loss functions agnostic) that extends standard classification to medical image segmentation. The key innovation is through designing a flexible loss framework and include a regularizer for controling the abstenion behavior. Experiments on CaDIS (cataract surgery) and DSAD (laparoscopic surgery) datasets with synthetic noise injection demonstrate consistent improvements over non-abstaining baselines.

**Strengths:**

A strength of this paper:

1. A novel application of dice segmentor which applies the standard classification design to segmentation problems

2. A unified loss design (equation 3) : it generalized abstention formulation with (1) informed regularization that incorporates estimated noise rates, (2) a flexible power-law-based auto-tuning algorithm for the abstention penalty parameter α

**Weaknesses:**

1. The mathematical clarity and justification of this paper should be detailed and improved
2. Limited baseline experiments on different backbone models.
3. Synthetic noise only: How well does the framework generalize to real annotation uncertainty where noise isn't cleanly separable?

**Detailed Comments:**

The paper presents a interesting and less discussed topic (abstention for noise-robust learning) and applied in medical image segmentation. The comments are mainly based on improving the mathematical clarity, justification, and baseline models (or ablation study).

**Justification Of Final Rating:**

Thanks for the rebuttal and revised paper. The revision was detailed and improved the sound of the paper, particulary by providing appendix and maths explanation. Final rating changes to weak accept after the rebuttal.

**Justification Of The Preliminary Rating:**

The paper presents an interesting and  less discussed topic (abstention for noise-robust learning) and applied in medical image segmentation. The framework proposed is innovative. The comments/improvements are mainly on mathematical clarity, justification, and baseline models (or ablation study).

**Questions To Address In The Rebuttal:**

1. The mathematical clarity and justification of this paper should be detailed and improved. For example, 3.3.2, the proposed Abstaining Dice segmenter, provide example of the loss functions based on the equation 3.

2. Limited baseline experiments on different backbone models. This paper uses U-Net as the backbone and compare the mIoU acorss different noise rate on CaDIS and DSAD datasets. How about other baseline models, and how are the performance?

3. Synthetic noise only: How well does the framework generalize to real annotation uncertainty where noise isn't cleanly separable? Simulated noise (morphological transformations + stochastic flipping) may not capture the complex, structured ambiguities in real clinical annotations

4. In Figure 2, add into captions that what b, c, h, w represent.

5. Based on 3.3.2, discuss more about why the class-wise abstention design are better ?

6.  The abstention penalty parameters (Appendix C.2, Table 3): It is confusing why and how the parameters L, α, γ, etc are selected.

---

> ### Author Response · Authors · 2026-01-24
>
> We thank the reviewer for the constructive feedback and for highlighting the novelty of the unified loss design and the application of abstention to segmentation. We have revised the manuscript to improve mathematical clarity and justify our design choices.
>
> **1. Mathematical Clarity (Q1)**
> We appreciate the request for explicit formulations. Based on the generalized framework in Equation 3, the specific loss functions are defined as follows:
> $$ \mathcal{L}\_{GAC}(x_j) = (1-p\_{k+1})\mathcal{L}\_{GCE}(x_j) + \alpha\left|\log\frac{1-\tilde\eta}{1-p\_{k+1}}\right|$$
> $$\mathcal{L}\_{SAC}(x_j) = (1-p\_{k+1})\mathcal{L}\_{SCE}(x_j) + \alpha\left|\log\frac{1-\tilde\eta}{1-p\_{k+1}}\right|$$
> $$\mathcal{L}\_{ADS}(x_j) = (1-p\_{k+1})\mathcal{L}\_{Dice}(x_j) + \alpha\left|\log\frac{1-\tilde\eta_c}{1-p\_{k+1}}\right|$$
> We deine the Dice loss as:
> $$\mathcal{L}\_{Dice}(x_j) = 1-\frac{2\sum\_{i=1}^N p_i t_i}{\sum\_{i=1}^N p_i+\sum\_{i=1}^N t_i}$$
> $x_j$ is the input image. $p_{k+1}$ is the predicted abstention probability for a pixel or class (pixel-wise or class-wise abstention). $\alpha$ is the scheduled abstention penalty. $\tilde\eta$ is the scalar dataset-level noise estimation. $\tilde\eta_c$ is the class-wise noise estimation vector of size $c$.
>
> **2. Baseline Models & Backbones (Q2)**
> We selected the U-Net architecture with a pretrained ResNet-50 backbone because it remains the *de facto* standard benchmark in medical image segmentation literature.
> Our primary goal was to validate a **universal loss framework**, not a new network architecture. By using the most widely adopted baseline, we isolate the performance gains to the *loss function itself* rather than architectural complexities. Since our framework operates strictly on the output/loss layer, it is theoretically compatible with any dense prediction architecture. We plan to expand our evaluation to architectures like DeepLabV3+ and FPN in future work, using this paper as the foundational proof of concept.
>
> **3. Synthetic vs. Real-World Noise (Q3)**
> We acknowledge that real-world clinical noise is often more complex than synthetic perturbations. However, the primary contribution of this paper is a **methodological framework**. To rigorously validate the mechanics of this framework (specifically the $\alpha$ auto-tuning dynamics and the regularization term), a controlled environment where the noise intensity $\eta$ is a known, tunable variable is scientifically necessary.
> Crucially, our noise protocol is not limited to random pixel flipping; it includes **structural noise** (morphological erosion/dilation) specifically designed to simulate the inter-rater variability and boundary uncertainty found in clinical annotations. We have added **Appendix D** with visualizations confirming that these perturbations create realistic boundary artifacts rather than just random noise.
>
> **4. Figure 2 Caption (Q4)**
> We have updated the caption in Figure 2 to explicitly define the dimensions: $b$ (batch size), $c$ (number of classes), and $h, w$ (spatial dimensions).
> Additionally, to avoid confusion with the input width $w$, we have renamed the output size of the adaptive pooling layer in the ADS head to $(s, s)$.
>
> **5. Justification for Class-wise Abstention (Q5)**
> The adoption of class-wise abstention for ADS was not an arbitrary choice, but a necessity enforced by the mathematical nature of the Dice loss.
> Abstention acts as an **inverse confidence score**: when confidence is low, the model “dampens” the loss to avoid overfitting. Standard abstention (DAC) operates pixel-wise, which aligns with pixel-wise losses like Cross Entropy. However, Dice is a **region-based** metric that computes a score *per class* across the entire image.
> In our initial experiments, combining Dice with pixel-wise abstention consistently failed, as the mechanisms operated at conflicting levels of granularity. The class-wise head resolves this by aligning the abstention mechanism with the loss function: it allows the model to down-weight the global Dice loss for specific, unreliable classes during training, rather than trying to filter individual pixels.
>
> **6. Hyperparameter Selection (Q6)**
> We have updated **Appendix C.2** to clarify our selection process. We utilized a two-stage optimization strategy:
> 1.  **Bayesian Search:** We used Weights & Biases to perform a broad Bayesian search to identify high-performing ranges for each parameter.
> 2.  **Grid Search:** We followed this with a fine-grained grid search over a set of interpretable, discrete values (e.g., 1.0, 1.5, 2.0) to select optimal values that are effective without being overfitted to a specific float value.
>
> **Clarification on $\alpha$:** It is important to distinguish between parameters. In our framework, $\alpha$ refers to the **abstention penalty weight**. In the specific context of the SCE baseline *only*, $\alpha$ refers to the weight of the Cross Entropy term (as defined in the original SCE paper).

---

### Official Review · Reviewer_BsCE · 2026-01-10

**Confidence:** 3
**Preliminary Rating:** 3
**Final Rating:** 4

**Summary:**

This paper addresses the issue of noisy labels in medical image segmentation by proposing a unified and modular abstention framework capable of enhancing the noise robustness of a diverse range of loss functions. The framework comprises of 2 keys components: an informed regularization component (a noise rate prior) to guide abstention behavior (i.e., abstaining on a percentage of pixels roughly equal to the estimated noise level in the dataset.) ; and a more flexible power-law-based auto-tuning algorithm for the abstention penalty. This power law creates a curriculum which ramps up the abstention penalty over time.

Experiments are performed on the CaDIS and DSAD datasets and involve integrating the abstention framework with three distinct noise functions to create noise-robust variants: GAC, SAC, and ADS.

**Strengths:**

1. The noise-informed regularization and the power-law auto-tuning are indeed innovative. The extension of abstention mechanisms to segmentation tasks is indeed novel.

2. At high noise levels, ADS outperforms standard Dice Loss by over 5% mIoU on CaDIS. The figures show that ADS produces clean, sharp masks even when trained on noisy data compared to baselines which produce artifacts.

3. The noise functions with the abstention framework-GAC, SAC and ADS- outperform the non abstaining noise functions (GCE, SCE, Dice).

4. The Class-based architecture introduced for adapting Dice Loss resolves the issue between pixel-level uncertainty and the global, region-based functionality of the Dice Loss.

5. The proposed abstention framework is plug-and-play allowing flexibility with taking different noise functions and making them more robust to noisy labels.

**Weaknesses:**

1. The noise rate prior relies on knowing the noise levels beforehand which is clinically impractical.

2. Most of the noise injection in the experiments is synthetic and not broadly representative enough of what one may encounter in practical situations. Using datasets with inter-rater variability or including synthetic artifacts such as incorrect boundaries around a vessel, missing small tumors would have made for a more interesting study.

3. For the Dice Loss, the framework switches to Class-wise abstention which may lead the model to abstain on an entire class versus specific noisy pixels. This may affect the segmentation robustness.

**Detailed Comments:**

no further comments to add

**Justification Of Final Rating:**

The authors have satisfactorily addressed the review comments and made changes to the paper that make it suitable for publication to the MIDL conference. After careful deliberation, I have decided to increase my original rating to 4.

**Justification Of The Preliminary Rating:**

The paper is indeed interesting and novel as it extends abstention frameworks to segmentation tasks in the presence of noisy labels. However, some questions remain regarding the clinical efficacy of the experiments and some of the results shared in this paper. I look forward to hearing back from the authors on the comments.

**Questions To Address In The Rebuttal:**

1. Would there be any experiments or methods which would dynamically estimate the noise levels ? (ex.: co-teaching, Confident Learning by Northcutt et al.)

2. Are there any additional experiments or results with more clinically realistic noise variations ?

3. Address point 3. in the "weaknesses" section

4. Clarify the inference part of the pipeline. Example: post abstention on a pixel, what is the clinical output - does the pixel class become void or is there an under-segmentation ?

5. Table 1 seems a bit unclear - each bracket seems to have multiple best results (highlighted in bold) ? Please clarify this.

6. Fig.3(a) - please explain why only ADS outperforms Dice (or is it CE represented by the dotted lines) at high noise levels versus GAC and SAC.

---

> ### Author Response · Authors · 2026-01-24
>
> We thank the reviewer for the positive assessment of our work, particularly regarding the novelty of the noise-informed regularization and power-law auto-tuning. We appreciate the constructive feedback and address your specific questions below.
>
> **1. Practicality of the Noise Rate Prior (Q1)**
> We acknowledge that relying on a pre-estimated noise rate is a practical limitation. To address this, we added **Section 5.1** and **Table 3** to the revised manuscript, analyzing the sensitivity of GAC, SAC, and ADS to mis-specified noise priors ($\tilde{\eta} \gg \eta$, $\tilde{\eta} \ll \eta$, and $\tilde{\eta}=0$).
> *   **Robustness:** SAC and GAC are highly resilient. Crucially, even with an uninformed prior ($\tilde{\eta}=0$), they consistently outperform their non-abstaining baselines.
> *   **ADS Specifics:** As shown in **Appendix D.1**, segmentation datasets often have high inter-class noise variance. Comparing Table 1 (Vector Prior $\tilde{\eta}_c$) to Table 3 (Scalar Prior $\tilde{\eta}$) confirms that ADS benefits from granular noise information. However, even with a scalar or uninformed prior, ADS still outperforms the baseline.
> *   **Dynamic Estimation:** We updated **Section 3.3.2** to recommend coupling our framework with dynamic estimation methods like Confident Learning (Northcutt et al.) or Beta Mixture Models to infer $\tilde{\eta}_c$ from data when manual estimation is infeasible.
>
> **2. Synthetic vs. Real-World Noise (Q2, Q3)**
> While synthetic noise is a proxy, it provides the controlled ground truth necessary to rigorously validate the framework's mechanics ($\alpha$-tuning, regularization). Our “structural noise” protocol (morphological erosion/dilation) specifically simulates the “incorrect boundaries” mentioned in your review. We added **Appendix D** with visualizations confirming that these perturbations create realistic boundary artifacts rather than just random noise. Validating on datasets with natural inter-rater variability is a priority for our future work.
>
> **3. Class-wise Abstention in ADS (Weakness 3)**
> Switching to class-wise abstention for ADS was not a choice, but a necessity enforced by the nature of the Dice loss.
> You can think of abstention as an **inverse confidence score** for the model's prediction. If the confidence in a sample is low (suggesting it is noisy), we saturate the loss so the model doesn't overfit. Since Dice produces a loss value *per class* (region-based) rather than per pixel, the abstention mechanism must match this granularity.
> In our early experiments, we attempted to combine Dice with pixel-wise abstention, but it consistently underperformed. Pixel-wise abstention actively worked against the class-wise objective of Dice, deteriorating performance. The class-wise head resolves this by allowing the model to down-weight the loss for specific, unreliable classes during training.
>
> **4. Inference Pipeline (Q4)**
> The abstention output is used **strictly for training**. For inference, the abstention channel (or head) is truncated. We simply calculate `argmax` on the remaining $k$ class channels to generate the segmentation map. A high abstention probability indicates low confidence during training, not a “void” label. Thus, the model produces standard, fully segmented masks without under-segmentation.
>
> **5. Table 1 Clarity (Q5)**
> We apologize for the confusion. We have updated the caption to explain the design and layout more clearly. In essence, Table 1 has 4 brackets/comparison groups. Each group pits a baseline loss function (e.g. Dice) against its abstaining counterpart (e.g. ADS). This allows us to contribute the improvements (or drops) in performance solely to abstention. The groups are [CE vs DAC vs IDAC], [GCE vs GAC], [SCE vs SAC], and [Dice vs ADS]. Each group will have **one optimal results per row/noise level**. This means that Table 1 has 4 optimal results per row/noise level, each one belonging to a group. While one can compare the results of all the losses against each other, this is not the intended way to analyze the results.
>
> **6. ADS Performance vs. GAC/SAC (Q6)**
> The performance difference stems from the dataset characteristics:
> *   **CaDIS:** Features large, contiguous objects (instruments, anatomy) with dense annotations. The **Dice Loss** thrives here because it directly optimizes overlap (IoU) for these large regions, allowing it (and ADS) to outperform pixel-wise CE-based losses.
> *   **DSAD:** Features sparse annotations (~18% foreground) and small, thin anatomical structures. In this regime, Dice is known to be unstable. **Pixel-wise losses** (like GCE/GAC) provide a more stable gradient signal for every pixel, allowing them to outperform Dice/ADS on this dataset.
> This validates the **modularity** of our framework: one can choose the base loss (GAC vs. ADS) that best fits the data density while retaining the robustness benefits of abstention.

---

### Official Review · Reviewer_KSx6 · 2026-01-17

**Confidence:** 4
**Preliminary Rating:** 4

**Summary:**

This paper extends the "learning to abstain" concept, originally established for classification via the Deep Abstaining Classifier (DAC), to the domain of robust medical image segmentation. The authors introduce a modular and loss-agnostic framework capable of wrapping base objectives, such as Dice or Generalized Cross Entropy, with an abstention mechanism. Unlike prior approaches that risk abstention collapse, this method employs an informed regularization term that actively tethers the abstention probability to a noise prior. Furthermore, the framework replaces heuristic scheduling with a simplified power-law auto-tuner for the penalty weight. The framework is instantiated as GAC, SAC, and the architecturally distinct Abstaining Dice Segmenter (ADS), which adapts the abstention head for class-wise losses. Validation on CaDIS and DSAD datasets indicates superior robustness against synthetic structural and semantic noise, with the proposed methods yielding significantly flatter degradation curves in high-noise regimes compared to non-abstaining baselines

**Strengths:**

1. **Effective extension from classification to segmentation.** The paper takes the *training-time abstention* idea (DAC) and shows it is not “just a classification trick,” but can be adapted to dense, pixel-level supervision where noise is spatially correlated and clinically consequential. The proposed framework explicitly targets this under-explored gap by designing abstention-aware objectives for medical image segmentation rather than only reusing classification recipes.

2. **A clean, loss-wrapper formulation validated across diverse robust losses.** The central design is a modular abstention wrapper that can be paired with multiple base objectives, and the paper demonstrates this concretely via GAC (with GCE), SAC (with SCE), and ADS (with Dice). This is especially persuasive because GCE and SCE are themselves established noisy-label robust losses, so the method is tested in combination with credible baselines rather than only CE.

3. **Clear implementation + solid experimental protocol across varied tasks.** The experimental setup is easy to audit (architecture, optimizer, schedules, seeds, noise injection, code) and the evaluation spans two distinct surgical segmentation datasets (CaDIS and DSAD) with multiple calibrated noise levels and 5-run mean±std reporting. The results show consistently flatter degradation curves for abstaining variants and strong high-noise gains (e.g., ADS leading Dice by +5.35 mIoU on CaDIS at 25% noise).

4. **Mathematical insights that fix a real failure mode (abstention collapse).** The informed regularizer explicitly anchors the abstention rate to a target noise estimate (\tilde{\eta}) instead of pushing abstention toward zero, and the power-law (\alpha) schedule provides a simple closed-form curriculum that avoids DAC’s stateful tuning logic. The appendix further supports the claim mechanistically: DAC collapses abstention after a spike, while the proposed GAC stabilizes near the intended rate (≈15% when noise is 15%).

**Weaknesses:**

1. **“Flatter degradation curves” is not quantified enough.**
   The paper repeatedly argues that abstaining variants have *flatter* performance drops as noise increases, mainly based on Figure 3 and wording in the text/caption.  A clearer, quantitative robustness metric (e.g., AUC of mIoU–noise curve, slope of a linear fit, or normalized drop rate ($\Delta\text{mIoU}/\Delta\eta$) with confidence intervals) would make this claim much stronger.

2. **Baselines are mostly “loss-only” comparisons; missing noisy-segmentation methods.**
   The experiments compare abstaining losses to non-abstaining baselines and other robust losses (GCE/SCE/Dice), but do not include representative noisy-label medical segmentation frameworks (e.g., pixel/image-level noise handling methods).  Adding 1–2 such baselines (or clearly explaining why they are excluded) would strengthen the positioning.

3. **Synthetic noise is not shown directly, so it is hard to judge realism and difficulty.**
   The paper describes structural (morphology) + semantic (label flipping) noise injection, but does not show examples of the corrupted labels themselves on top of the original images.  Showing “original image + GT + noisy label + difference/overlay” would make the noise setup much more credible.

4. **Table 1 highlighting rules are confusing.**
   Table 1 says abstaining losses are gray, proposed losses are bold, and best-in-bracket is also bold—so bold has two meanings.  This can confuse readers about what is “proposed” vs what is simply “best.”

5. **The method depends on a noise-rate hyperparameter ($\tilde{\eta}$), but sensitivity is not tested.**
   The framework uses an informed regularizer that needs a pre-estimated noise rate, and the conclusion also acknowledges this limitation.  A small sensitivity study (e.g., ($\tilde{\eta}\pm 5%$)) would help evaluate how practical this is when the true noise rate is unknown.

**Detailed Comments:**

1. **Clarify Table 1 formatting.**
   Please adjust the caption/legend so it is unambiguous: gray background = abstention, and use a different mark for “proposed” (e.g., underline or a ★), while reserving **bold only for best** results.

2. **Define “calibrated noise level” precisely.**
   The paper reports 5–25% (CaDIS) and 3–15% (DSAD) corruption, but it is not fully clear whether this percentage is computed as fraction of pixels changed, per-class pixel fraction, or something else.  A one-sentence definition (or short pseudocode) would improve reproducibility.

3. **Show the noisy labels visually (not only model outputs).**
   Add a figure panel for the corrupted label next to GT, ideally with label boundaries overlaid on the original image. This helps readers interpret how severe “15%” or “25%” looks.

4. **State how ($\tilde{\eta}$) (and ($\tilde{\eta}_c$) for ADS) is set in experiments.**
   Since noise is synthetic, you can compute the true corruption rate exactly; please state whether ($\tilde{\eta}$) uses that exact value, and how class-wise rates are computed for ADS.

**Justification Of The Preliminary Rating:**

I vote weak accept because the paper makes a clean, practical extension of abstention-based learning to medical image segmentation, with a simple loss-wrapper formulation and consistent improvements across two datasets under multiple noise levels. The main gaps (limited noisy-segmentation baselines, no direct visualization of the injected noisy labels, and “flatter robustness” argued mostly qualitatively) seem fixable and do not outweigh the solid technical contribution and clear, reproducible implementation.

**Questions To Address In The Rebuttal:**

1. **Can you show examples of the synthetic noisy labels overlaid on the raw images?**
   Please include “image + GT + noisy label (overlay) + difference map” for multiple noise levels. This would help validate that the synthetic noise is realistic and sufficiently challenging.

2. **Can you support the “flatter degradation” claim with a quantitative robustness metric?**
   For example: AUC of mIoU–noise curve, slope of the degradation line, or normalized performance drop, with mean±CI over 5 seeds (and ideally a simple significance test).

3. **Can you add (or at least discuss) stronger noisy-segmentation baselines?**
   Including one representative method such as PINT (or a similar pixel/image-level noisy-label segmentation framework) would clarify where abstention stands relative to segmentation-focused noisy-label solutions.

4. **How sensitive is performance to ($\tilde{\eta}$) being wrong?**
   A small sweep where ($\tilde{\eta}$) is under/over-estimated would help judge practicality when the true noise rate is unknown.

---

> ### Author Response · Authors · 2026-01-24
>
> We thank the reviewer for the positive assessment and for recognizing the novelty of extending abstention to segmentation, the clean modular formulation, and the solid experimental protocol. We appreciate the constructive feedback and have incorporated your suggestions into the revised manuscript.
>
> **1. Visualizing Noisy Labels (Q1)**
> We agree that visualizing the synthetic noise is crucial for judging realism. We have added **Appendix D** to the revised manuscript, which contains two new figures (Figures 8 and 9) showcasing the “quad-view” requested: (1) Original Image, (2) Ground Truth, (3) Noisy Mask Overlaid on Original Image, and (4) Difference Map.
> These visualizations clearly depict that our noise generation protocol introduces not only random semantic flips but also challenging structural artifacts along object boundaries (e.g., erosion/dilation of surgical tools in CaDIS and anatomical structures in DSAD), confirming that the noise model mimics realistic inter-rater variability rather than simple random noise.
>
> **2. Quantifying “Flatter Degradation” (Q2)**
> To rigorously support the visual trend in Figure 3, we calculated the **Normalized Performance Drop Rate** ($\Delta mIoU / \Delta \eta$) across the 5 random seeds with confidence intervals (Mean $\pm$ 95\% CI).
> We performed a paired t-test between our methods and their baselines and have updated **Table 2** in the revised manuscript to report these metrics. Statistically significant improvements ($p < 0.05$) are now marked with a dagger ($\dagger$).
> For example, on the CaDIS dataset, ADS demonstrated a significantly lower degradation rate than the Dice baseline (ADS: $0.426 \pm 0.036$ vs. Dice: $0.619 \pm 0.079$, $p < 0.001$). This provides quantitative statistical evidence that the abstention mechanism effectively dampens the impact of increasing label noise.
>
> **3. Comparison to Noisy-Segmentation Baselines (Q3)**
> We have already discussed several segmentation-focused noise mitigation solutions in the **Related Work (Section 2)**, including PINT.
> Regarding the comparison: We respectfully argue that our contribution—a modular loss function framework—is **orthogonal** rather than competitive to complex pipelines like PINT. Our goal was to demonstrate that *abstention alone*, implemented via a lightweight loss function change, provides significant robustness without the architectural overhead of multi-stage pipelines.
> Theoretically, our framework can be deployed **alongside** these methods. Since abstention only requires a modification to the output layer (adding an abstention channel or head) and the loss function, it can enhance existing noise-correction pipelines rather than replacing them. We are excited for other researchers to experiment with abstention and see how it can improve more complex noise mitigation frameworks.
>
> **4. Sensitivity to $\tilde{\eta}$ (Q4)**
> We have added a new **Subsection 5.1 (Sensitivity to Noise Rate Estimation)** and **Table 3** to address this practical concern directly.
> We performed a comprehensive ablation study on GAC, SAC, and ADS, testing each loss with four mis-specified noise priors (two over-estimated and two under-estimated, including uninformed $\tilde{\eta}=0$).
> *   **SAC:** Demonstrated remarkable stability, with performance remaining virtually unchanged regardless of the prior.
> *   **GAC:** Showed slight sensitivity but remained robust; even with an uninformed prior ($\tilde{\eta}=0$), it significantly outperformed the baseline.
> *   **ADS:** We observed that ADS is more sensitive to the *granularity* of the prior (scalar vs. vector) than the *magnitude*. In this sensitivity study (using scalar $\tilde{\eta}$), ADS performance dropped compared to our main results in Table 1 (using vector $\tilde{\eta}_c$), confirming the benefit of our class-wise architecture. However, even with a scalar prior—or indeed, an uninformed prior ($\tilde{\eta}=0$)—ADS still outperformed the non-abstaining Dice baseline.
>
> This confirms that while an accurate prior is beneficial, the framework is robust and safe to use even when the true noise rate is unknown.

---

### Author Rebuttal · Authors · 2026-01-24

**Rebuttal:**

We thank the reviewers for their constructive feedback, which has significantly strengthened the manuscript. We have revised the paper to improve clarity, rigorous validation, and reproducibility. The key changes are as follows:

1.  **Quantitative Robustness Metrics:** We added a new quantitative analysis in **Section 5** using the **Normalized Performance Drop Rate** ($\Delta mIoU / \Delta \eta$). We performed paired t-tests to validate the statistical significance of our improvements, marking significant gains ($p < 0.05$) in **Table 2**. This mathematically substantiates the “flatter degradation” claims observed in the plots.

2.  **Sensitivity Analysis & Practicality:** We introduced a new **Subsection 5.1** and **Table 3** to evaluate the framework's sensitivity to the noise rate prior ($\tilde{\eta}$). We conducted an ablation study mis-specifying $\tilde{\eta}$ (under/over-estimation and uninformed $\tilde{\eta}=0$). The results demonstrate high robustness for GAC/SAC and confirm that ADS remains effective even without an informed prior, though it benefits from granular class-wise information.

3.  **Visualization of Realistic Noise:** To address concerns about synthetic noise realism, we added **Appendix D** containing detailed visualizations (Original Image, Ground Truth, Noisy Overlay, Difference Map) for both datasets. These figures confirm that our noise protocol introduces challenging structural artifacts along boundaries, mimicking inter-rater variability.

4.  **Clarifications & Formalizations:**
    *   We added precise mathematical definitions for the noise rate $\eta$ and class-wise noise vector $\tilde{\eta}_c$ in **Section 4** and **Appendix D.1**.
    *   We expanded **Appendix C.2** to detail our two-stage hyperparameter optimization strategy (Bayesian search followed by grid refinement).
    *   We updated the **Table 1** caption and formatting to clearly distinguish between comparative groups, resolving ambiguity about “best results.”

5. **Refined ADS Methodology:** We updated **Section 3.3.2** to clarify that ADS accepts scalar noise rates for flexibility. We discussed the performance trade-offs validated in our sensitivity study and recommended dynamic noise estimation (e.g., Confident Learning) as a robust alternative.

**Supporting Material:**

/attachment/f17addfe46ce33ab8907f18567582838f738e8e0.pdf

---

### Meta-Review · Area_Chair_AbYZ · 2026-02-07

**Recommendation:** Accept (Poster)
**Confidence:** 4

**Metareview:**

This paper presents a timely and well-crafted contribution that successfully extends the concept of training-time abstention from classification to the critical domain of medical image segmentation. The authors translate the core idea (teaching a model to selectively ignore potentially corrupted pixels during training) into a clean, modular, and loss-agnostic framework. The work is validated through rigorous experiments on two distinct medical segmentation datasets, demonstrating consistent and significant improvements in noise robustness.

The initial concerns regarding quantitative validation, sensitivity, and clarity were adequately addressed in the revision.

A minor point: the variable $t_i$ in Equation 1 is not defined (This might be obvious, but it’s not good practice to leave readers guessing). The authors should add a brief description.

It presents a solid technical contribution that will benefit from direct discussion with the community, allowing for deeper exchanges on implementation details and potential applications to real-world noisy datasets. For these reasons, I recommend this paper be accepted.

---

### Decision · Program_Chairs · 2026-02-13

Accept (Poster)